# OpenReview forum: "Acoustic Interference: A New Paradigm Weaponizing Acoustic Latent Semantic for Universal Jailbreak against Large Audio Language Models"
_ICML.cc/2026/Conference — ICML 2026 regular_

### Official Review · Reviewer_zZyL · 2026-03-02

**Soundness:** 3
**Presentation:** 3
**Significance:** 2
**Originality:** 3
**Overall Recommendation:** 4
**Confidence:** 4

**Summary:**

The paper introduces "Acoustic Interference", a new jailbreak notion against Large Audio Language Models (LALMs). The core idea is that benign audio that contains no malicious semantic content, but with specific paralinguistic features termed Acoustic Latent Semantics (ALS), can serve as a universal trigger that drifts LALM inference away from safety-aligned states. The authors construct an ALS arsenal by mining Bark's generative latent space, rank interference cadidates offline, and propose the Acoustic Interference Attack (AIA), which paris standard jailbreak text with top-ranked benign audio. They evaluate on 10 LALMs across five datasets, claiming SOTA attack success rates. Interpretability analyses via logit shifts, latent projects, and causal activation patching aim to explain the mechanism. A 12D labeling system characterizes which paralinguistic patterns are most effective.

**Compliance With Llm Reviewing Policy:**

Affirmed.

**Key Questions For Authors:**

**Questions**

1. The entire ALS arsenal comes from Bark. How do you know this isn't just exploiting Bark-specific audio artifacts rather than discovering universal paralinguistic features?

2. The paper claims that the audio is 'benign' and 'semantically neutral.' How rigorously is this verified?

3. Why does ALS suppress strong attacks? The bi-directional interference is the paper's most fascinating finding, but the mechanistic explanation is essentially hand-waving.

4. ASR-R gives you 74% but ASR-M gives you 100% on Qwen2.5-Omni. Which number should we believe?

5. The ranking uses λ_weak=3, λ_medium=2, λ_strong=1. How sensitive is performance to these weights?

**Potential Missing Citations**
- SACRED-Bench appears prominently in Table 2 with numerical results but is absent from the reference list.
- Relevant defense papers: hope the reviewers check existing defense mechanisms such as SafeEar (https://arxiv.org/pdf/2409.09272), SEA (https://arxiv.org/pdf/2502.12562)
- JailbreakLens (https://arxiv.org/pdf/2411.11114): performs representation and circuit analysis of jailbreak mechanisms. Given the paper's interpretability claims, this should be discussed.
- Work on boosting transferability via acoustic representation optimization  (https://arxiv.org/pdf/2503.19591) s relevant to understanding why Bark-generated ALS transfers across different LALM

**Limitations:**

**1. Surrogate Model Bias and Generalizability of ALS Arsenal**
The entire ALS arsenal is constructed from a single generative model (Bark). This raises a fundamental question: are the discovered ALS patterns truly universal properties that exploit cross-modal alignment vulnerabilities, or are they artifacts of Bark's specific generative distribution? The paper does not investigate:
- Whether ALS arsenals constructed from different TTS systems (e.g., VALL-E, XTTS, Tortoise) yield similar or different effective patterns.
- Whether the Bark-specific generation artifacts (glitch/artifacts dimension) are driving results via encoder confusion rather than genuine paralinguistic interference.
- The overlap between Bark's latent space and the audio encoders used by target LALMs.

This single-surrogate design significantly limits the strength of the "acoustic latent semantics" claim. If Bark's specific audio characteristics (compression artifacts, codec behavior) are what actually drive the attack, the paradigm story is much weaker.

**2. Unfair Baseline Comparison in Table 2**
Table 2 is the centerpiece comparison, but several issues could undermine its conclusions:
- Heterogeneous evaluation conditions: Different baselines use different datasets (some ◦ full, some ✓ partial), different model subsets, and different query budgets. Direct ASR comparison across rows is misleading when the underlying evaluation protocols differ substantially.
Missing baselines from Table 2 references: SACRED-Bench appears in Table 2 with detailed numbers but is not included in the reference list — this is a citation error that must be corrected.
- Cherry-picked model coverage: AIA is tested on a specific set of models, while baselines are reported on partially overlapping but different model sets. Many cells are empty ("–"), making it impossible to do controlled pairwise comparisons.
Red vs. black numbers: The paper notes some numbers come from original papers (black) vs. JALMBench (red), but doesn't clarify whether these use identical evaluation pipelines. Cross-benchmark ASR numbers are not directly comparable.

A fairer comparison would re-implement key baselines under identical conditions, or at minimum, clearly discuss these limitations.

**3. Evaluation Metric Concerns**
The introduction of ASR-M alongside ASR-R, while motivated by a reasonable observation (that non-prefix-optimizing attacks may be penalized by refusal string filters), could introduce ambiguity:
- ASR-M disables refusal filters and falls back to GPT-4o-mini when GPT-4o "fails to judge three times." What constitutes a GPT-4o "failure"? API errors? Refusals to evaluate harmful content? Low-confidence scores? This is underspecified.
- The gap between ASR-R and ASR-M is enormous in some cases (e.g., Qwen2.5-Omni: 74.47 → 100.00 on JBB). When the two metrics diverge this dramatically, it's unclear which one to trust, and the paper doesn't adequately reconcile this.
- The human evaluation (Table 4) is limited to mean score comparison on a single dataset with only 10 volunteers. There is no inter-annotator agreement reported (e.g., Krippendorff's alpha or Fleiss' kappa), and the sample size is too small for robust statistical conclusions.

**4. Incomplete Analysis of the Bi-Directional Effect:**
The paper's most interesting finding — that ALS suppresses already-strong attacks while amplifying weak ones — is presented as an "interference" phenomenon but not deeply explained. Section 2.2 acknowledges this but defers to Section 4.1, which focuses mainly on the amplification case. The suppression effect on strong attacks deserves much more attention: *If ALS causes a general drift from safety subspace, why would it hurt already-successful attacks?* A drift should be additive in the same direction.

**Strengths And Weaknesses:**

1. **Comprehensive experimental coverage**:  Testing across 10 LALMs including proprietary and 3 standard benchmarks. The comparison with 9 baselines in table 2, though imperfect, provides useful context.

2. **Novelty of the attack**: The idea that attack payload can be decoupled from audio modality entirely with audio serving only as an alignment interference mechanism rather than a content carrier is quite insightful. This challenges a strong implicit assumption in the field of audio safety that audio jailbreaks must encode malicious content and opens a genuinely new research direction. The separation of existing paradigms into semantic optimization, acoustic control, and additive perturbation is clear and useful.

3. **Multi-Faceted Interpretability Analysis**: The three-pronged mechanistic investigation goes beyond surface-level evaluation. The causal patching experiment (Section 4.1, Figure 5c) is particularly valuable — demonstrating that patching audio activations into text-only streams causally induces unsafe behavior, while the reverse restores safety. This provides stronger evidence than correlational analyses alone.

---

> ### Author Rebuttal · Authors · 2026-03-30
>
> Thank you for your detailed reviews. We really appreciate you for encouraging our comprehensiveness, novelty, and interpretability. We hope the following response could help address your concerns :)
>
> ---
>
> **Q1&L1: Surrogate bias and generalizability of ALS**
>
> Please kindly refer to "W1" in the response to Reviewer fyAf. As there is a similar question, we combine all the points from both of you. Thank you!
>
> ---
>
> **Q2: Verifying 'benign' and 'semantically neutral' audio**
>
> Thanks for your detailed question. Here, 'semantically neutral' means no malicious semantic content in the audio. This is because there are only three kinds of content, as detailed in Fig 3 and Sec 2.2, all of which are just neutral instructions. 'Benign' means no malicious, structured signal injection into the audio, such as additive adversarial noise. This is also straightforward, as all ALS are originally from unconditional sampling from a natural audio generative space, without any human crafting. This is also double-comfirmed by humans, where most of our interference audios are not distinguishable from typical TTS results. As promised in the manuscript, we would open both the code and the ALS arsenal, so that the public could also, hopefully, double-confirm this.
>
> ---
>
> **Q3&L4: Bi-directional interference**
>
> Thank you for your interest in this finding. Actually, we believe the drift, different from conventional ones, is not directional regarding the whole input space. This is quite natural because our interference audio does not contain any deliberate information to pursue a directional drift. That is why we position it as "interference" instead of something like "adversary" or "degradation". It is most probable that, the drift direction tends towards a Gaussian w.r.t. all cases, and only when we divide such cases into originally strong to weak sets, different directional patterns could be empirically captured. Then why do the patterns show a drift from safety subspace in weaker sets, while showing the opposite in stronger sets? We think this is because jailbreak is based on very specific "outliers" (it may not be that weird in distribution but should contain some rare and precise features). Therefore, an additional interference is likely to disrupt the originally precisely triggered malicious activity. On the contrary, in the original failure cases, the interference means a greater chance of escaping from the safety-aligned inference path, thus statistically showing a drift from safety subspace in general.
>
> ---
>
> **Q4&L3: ASR-R vs. ASR-M**
>
> Thank you for the detailed question. We appreciate that you think the motivation of ASR-M is reasonable. Here, "GPT-4o failure" means it refuses to evaluate the response, probably because it contains harmful content that violates safety alignment. As suggested in Sec 3 and Table 3, when ASR-R and ASR-M significantly diverge, the newly introduced ASR-M is more trustworthy, because it fits better with three external evaluations: two safeguard models, HarmBench classifier and Llama Guard 3, plus a human scoring. While acknowledging that the human evaluation itself may not be that robust, together with the external safeguard models, provided that all three results are highly consistent with each other and ASR-M, we respectfully believe this is sufficiently strong evidence to support ASR-M.
>
> ---
>
> **Q5: Ranking weights**
>
> Thanks for noticing this detail. We acknowledge that the weight settings are somewhat arbitrary, but this is because our performance is not sensitive to the ranking. In other words, the whole ALS arsenal is effective, rather than only the top 30 (still, a reasonable ranking would be beneficial). Table 5 in Appendix C provides evidence for this statement.
>
> ---
>
> **Q6: Potential missing citations**
>
> Thank you for pointing out the missing reference and suggesting the relevant works. We are glad to refine our manuscript and add discussions accordingly.
>
> ---
>
> **L2: Unfair comparison in Table 2**
>
> Thank you for the comment. Regarding the heterogeneous evaluation conditions, please kindly refer to "W1&Q2" in the response to Reviewer qgiu, where we explain why we respectfully believe the current manner is reasonable, and provide additional experiments to enhance our evaluation.
>
> Besides, we would like to respectfully clarify that there is no cherry-picking in our model coverage. We cover all models appearing in baselines (even only once), except for the Gemini family, where we cover the SOTA 3-pro and 2.5-pro, getting results better than baselines on weaker versions, so we sincerely believe it is not necessary to further cover them. For other empty cells, another reason to leave them is to reveal the fact that the corresponding models are not covered by certain baselines, demonstrating our advantage in coverage scope.
>
> Finally, regarding red vs. black numbers, we would add a detailed explanation in our manuscript. Sorry we have run out of space till here, thus could not directly provide it.

---

> > ### Author Rebuttal · Reviewer_zZyL · 2026-04-02
> >
> > We thank the authors for the detailed response and I remain my score.

---

> > > ### Author Response · Authors · 2026-04-02
> > >
> > > Thank you for your timely feedback! We appreciate your positive rating.

---

### Official Review · Reviewer_qgiu · 2026-03-05

**Soundness:** 3
**Presentation:** 3
**Significance:** 2
**Originality:** 3
**Overall Recommendation:** 4
**Confidence:** 3

**Summary:**

This work studies universal jailbreaks against Large Audio Language Models (ALMs) by proposing a new paradigm and an Acoustic Interference Attack (AIA), where audio shifts from content injection to interfering with safety alignment. Rather than embedding malicious instructions in spoken audio, AIA searches for audio clips that disrupt an ALM’s safety behavior, while the malicious instructions remain in the text prompt. The method first constructs an Acoustic Latent Semantic Arsenal using TTS (Bark) and audio–text models such as CLAP and WavLM to capture high-level semantic attributes. These audio clips are then used to perform query-based black-box attacks on ALMs. Experiments demonstrate strong effectiveness.

**Compliance With Llm Reviewing Policy:**

Affirmed.

**Final Justification:**

The authors have provided additional results for fair comparison with baseline methods and clarification in the settings. Considering these can be added to the revision, I think this work could be accepted.

**Key Questions For Authors:**

- If the objective is to elicit harmful content, what advantages does audio offer over vision or text-only attacks? In particular, the paper should compare against stronger text-based jailbreak baselines (not only naive “plain malicious instruction” prompts as in Table 1), to justify why adding an audio modality increases attack capability.

- Under a unified experimental protocol (same target models, prompts/tasks, safety settings, and query budgets), what is the benefit of the proposed method relative to prior baselines?

**Limitations:**

Given the topic of this work, it would be beneficial to include a dedicated discussion of the potential negative societal impacts, rather than relying solely on a generic template statement.

**Strengths And Weaknesses:**

**Strengths**

*Soundness:* The proposed method appears technically sound. The experimental design broadly matches the research aims, and the claimed new paradigm that vulnerabilities arise from safety alignment interference rather than content injection is supported by the reported results.

*Originality:* The core insight is novel and valuable: using the audio modality as interference while keeping malicious intent in the text prompt is important for jailbreak research.

---

**Weakness**

*Soundness*

- Although the experimental setup is generally reasonable, the baseline comparisons may not be fair. If I understand correctly, the ASR values reported in Table 2 are taken from original papers or JALMBench, rather than being reproduced under a unified protocol. This raises the possibility that different approaches were evaluated under different settings. Such mismatch can affect ASR and query efficiency, making the comparison difficult to interpret. Ideally, all methods (including the proposed one) should be evaluated under identical or at least tightly matched settings.

*Presentation*
- The proposed method is essentially a query-based attack rather than a universal attack. The paper should clearly define and distinguish universal vs. query-based attacks, and explain in what sense the proposed approach is “universal”.
- The font in Table 2 is too small, reducing readability. The use of colors and boldface lacks explanation may confuse readers.
- SACRED-Bench appears to be missing a reference (or is not properly cited).

*Significance*

- The motivation for using audio as the jailbreak modality is not fully convincing. Other modalities (e.g., vision) or even purely text-based attacks may achieve similar outcomes, yet the paper does not provide cross-modality comparisons or a clear argument for why audio is uniquely important.
- The empirical results also do not clearly demonstrate superiority over existing methods. For example, SACRED-Bench appears to achieve comparable ASR with a lower query budget, which weakens the case for practical advantage.

---

> ### Author Rebuttal · Authors · 2026-03-30
>
> Thank you for your detailed comments and encouragement on our technical soundness and originality. Below we'd like to respond to each point of your concerns :)
>
> ---
>
> **W1&Q2: Baseline comparison under unified protocol**
>
> Yes, the understanding is totally correct, and we fully understand your point. But if we, on the other hand, reproduce them, find some worse results, and report, then there could be another concern like: "Why didn't we refer to the original results? Whether we deliberately degrade baselines?" It seems our community has not reached a consensus on the best practice. In our case, we tend to the former because:
>
> - Most of the baselines (7 in 9) are instance-specific, which are not directly competitive to universal methods like ours. For simple reference rather than strict comparison, we believe the original/JALMBench results would be qualified.
>
> - Jailbreak is quite a practical task. So perhaps, compared with unifying the settings of all papers, it might be better to believe that their authors have found their own best practices and are willing to bear the costs (like the specific query time).
>
> Still, we are glad to provide a reproduction of the two universal baselines under a unified protocol (Qwen2.5-Omni/GPT-4o-Audio, JBB prompts, 12 query times) for your kind reference. AMSE is open-source, where we directly use their toolbox, while AudioJailbreak is not, where we try to reproduce a conceptually similar method.
>
> ||AudioJailbreak|AMSE|AIA (Ours)|
> |:-:|:-:|:-:|:-:|
> |Qwen2.5-Omni| 6.82|37.78|100.00|
> |GPT-4o-Audio|4.26|6.82|75.61|
>
> ---
>
> **W2: Query-based attack vs. universal attack**
>
> Thank you for noticing this detail. We understand your meaning, but respectfully disagree. Universal means directly applicable without any instance-specific operations, like depending on specific attack content to initialize the query or relying on model feedback to optimize the attack in the query loop. The latter is the most important feature of query-based attacks. They utilize query results from previous turns to optimize subsequent queries, which is not adopted by our work. The specific query time, on the other hand, is not directly relevant to the universal. For instance, among our baselines, Multi-AudioJail only queries once, but it initializes based on specific attack content, thus is not universal. In contrast, AMSE queries 18-times on average, but it still meets the requirement of universal. We are glad to make this clearer in our manuscript.
>
> ---
>
> **W3&W4&L1: Table readability & missing reference & potential impact**
>
> Thank you for your kind suggestions. We would accordingly refine our manuscript.
>
> ---
>
> **W5&Q1: Motivation for audio jailbreak & comparison with vision/text-only attacks**
>
> Thank you for the question. First, vision jailbreaks would not impact our motivation, because there are non-omni LALMs like those in our experiment, Qwen2-Audio, Kimi-Audio, and MiMo-Audio, where vision input is invalid. When the real-world target is such a model, only audio/text attacks are applicable.
>
> Then, the unique importance of audio is its generalizable effectiveness in enhancing text-only attacks under the newly defined audio interference paradigm. We would like to respectfully clarify that our text-only baselines are not naive "plain malicious instructions". In Table 1, JBB text is from PAIR, one of the most recognized SOTA semantic jailbreaks, which employs advanced semantic-level strategies, like "role-playing, logical appeal, and authority endorsement". Also, WildJailbreak "mines in-the-wild user-chatbot interactions to discover novel jailbreak tactics and composes them for systematic jailbreak". Both of them are representative of recent advances in text-only attacks.
>
> Thus, Table 1 actually demonstrates the consistent effectiveness of our method in enhancing advanced text jailbreaks. It gives us confidence to believe, given unseen jailbreak texts, our method is still expected to consistently enhance them, rather than achieve similar outcomes. In other words, we would not compete with stronger text attacks but cooperate instead.
>
> ---
>
> **W6: Superiority over existing methods**
>
> Thanks for the comment. However, we respectfully disagree. First, instance-specific attacks are not directly competitive to universal ones, especially in real-world practice, because they need query optimization and online TTS, both are time- and resource-consuming. Second, query cost is not simply query times but multiplied by query length. In LALM, the cost is mostly on audio. Our audio length per query is 3~4 words on average like "sure, here is", while all beselines need to carry malicious content. Taking SACRED-Bench for example, as long as its audio query is about 6x longer than ours (which is almost certain), its query cost would actually be higher. Finally, even directly comparing our method with the best baseline, SACRED-Bench, the result would be 85.62 vs. 79.89 on average, demonstrating a significant advantage of ours.

---

> > ### Author Rebuttal · Reviewer_qgiu · 2026-04-02
> >
> > Thank you to the authors for the detailed responses.
> >
> > W1 & Q2: I understand the authors’ point regarding the difficulty of reporting results in a fully fair manner, and I agree that achieving consensus on this issue in this field may be challenging. The additional experiments are helpful. I would encourage the authors to extend this evaluation to all methods so that the comparison with the baselines is more consistent, while still retaining the originally reported numbers for reference.
> >
> > W2: I understand that the universal attack is intended to be applicable across different jailbreak targets. If I understand correctly, the proposed method remains universal even after querying the victim models. However, the current presentation is somewhat unclear on this point. I would suggest clarifying this in the revision, possibly with reference to [1,2,3], and explicitly distinguishing the proposed method from “transfer”-based universal attacks, which are closer to a zero-shot setting and do not rely on multiple queries.
> >
> > I hope the authors can incorporate these improvements in the revision. With clearer clarification of the experimental setting, a more precise positioning relative to related work, and additional results to enable fair comparison with baseline methods, I will increase my score to 4. Good luck!
> >
> > [1] Moosavi-Dezfooli, S. M., Fawzi, A., Fawzi, O., & Frossard, P. (2017). Universal adversarial perturbations. In Proceedings of the IEEE conference on computer vision and pattern recognition (pp. 1765-1773).
> >
> > [2] Zou, A., Wang, Z., Carlini, N., Nasr, M., Kolter, J. Z., & Fredrikson, M. (2023). Universal and transferable adversarial attacks on aligned language models. arXiv preprint arXiv:2307.15043.
> >
> > [3] Cui, K., Li, Y., Wu, Y., Ma, X., Erfani, S., Leckie, C., & Huang, H. (2026). Toward Universal and Transferable Jailbreak Attacks on Vision-Language Models. arXiv preprint arXiv:2602.01025.

---

> > > ### Author Response · Authors · 2026-04-02
> > >
> > > Thank you very much for your detailed feedback and kind suggestions regarding more solid evaluation and more precise positioning, along with the detailed recommendation on the references. We really appreciate it. Though we are not allowed to change the PDF during the rebuttal, we promise to incorporate all these improvements in the future revision. Thank you so much for increasing the score!

---

### Official Review · Reviewer_dpi2 · 2026-03-09

**Soundness:** 3
**Presentation:** 3
**Significance:** 3
**Originality:** 3
**Overall Recommendation:** 4
**Confidence:** 2

**Summary:**

The paper addresses the fundamental vulnerability of cross-modal safety alignment in LALMs by introducing a new attack, called the Acoustic Interference Attack (AIA).
Unlike previous research, the authors decoupled the roles of two modalities, leaving the malicious instructions of the attack entirely to the text, while the audio, containing something completely benign like "Sure, here is~", only serves to disrupt the safety filter.
The paper shows a higher attack success rate than existing methods.

**Compliance With Llm Reviewing Policy:**

Affirmed.

**Final Justification:**

the authors' detailed response has resolved my concerns, and I maintain my positive assessment of the paper.

**Key Questions For Authors:**

- Can a truly neutral sentence that doesn't induce model compliance also lead to a successful attack? (e.g. I'm happy to ~)
- What is the actual quality of the 'Invalid' responses?
- Did you conduct an ablation study to strictly isolate the effects of synthetic noise (glitches/artifacts) from pure acoustic features (ALS)?

**Limitations:**

yes

**Strengths And Weaknesses:**

### Strength
- The paper achieves a high attack success rate without requiring a separate audio optimization process for malicious audio generation, unlike previous studies.
- The paper discovers a new vulnerability showing that not only malicious semantics but also interference from non-linguistic acoustic features unrelated to the content can cause models to deviate from their reasoning trajectory, thereby disabling safety filters.
### Weakness
- The paper used only 3 fixed sentences (e.g. "Sure, here is,", "Below is an instruction...", ...). In particular, "Sure, here is" is a phrase optimized for model compliance in the previous jailbreak research. So, the conclusion that the attack was successful solely due to 'acoustic interference' may have been contaminated by a controlled variable
- It is unclear whether the trigger for breaching the model's safety net is synthetic noise or acoustic features.

---

> ### Author Rebuttal · Authors · 2026-03-30
>
> Thank you for your comments and for encouraging our effectiveness and novelty. We hope the following response could help address your concerns :)
>
> ---
>
> **W1&Q1: Sentences for acoustic interference**
>
> Thank you for noticing this detail. We fully understand your point. However, just as you say, "sure, here is" is optimized for model compliance in previous jailbreak research, but the other two sentences are not (e.g., “below is an instruction that describes a task” is a standard system prompt for LLM instruction tuning, which we respectfully believe is "truly neutral"). We have actually compared these three sentences in an internal experiment before (as the result posted below), and did not see a significant difference in effectiveness. Thanks for your kind reminder, we now realize that it would be better to show this internal result in our manuscript as well. Still, we are also glad to add an experiment with the suggested sentence "I'm happy to" (on JBB and Qwen2.5-Omni/GPT-4o-Audio), with the ASR result illustrated below for your kind reference. Based on all these results, we sincerely believe the possibility that the successful attack is due to sentence content rather than acoustic interference could be ruled out.
>
> |              | Our Sentence 1 | Our Sentence 2 | Our Sentence 3 | "I'm happy to" |
> |--------------|:--------------:|:--------------:|:--------------:|:--------------:|
> | Qwen2.5-Omni |     100.00     |     100.00     |      98.04     |      98.31     |
> | GPT-4o-Audio |      74.36     |      73.68     |      72.73     |      73.81     |
>
> ---
>
> **W2&Q3: Glitches/artifacts vs. pure ALS**
>
> Thank you for the question. First, we would like to respectfully clarify that glitches/artifacts are not isolated from ALS. Instead, as detailed in Sec 4.2, Fig 6, Fig 7, and Appendix F, it is one of the 12 dimensions of ALS. Then, whether glitches/artifacts, rather than other acoustic features, trigger the breach of the safety alignment? We sincerely believe the strict ablation for each of the 12 dimensions in Sec 4.2 has eliminated such a possibility. As shown in Fig 6 and 7, the top-3 effective acoustic features are respectively "Persona-Pitch", "Delivery-Valence", and "Delivery-Speed", while "Signal-Glitch/Artifacts" only shows a moderate level of influence.
>
> ---
>
> **Q2: Actual quality of invalid responses**
>
> Thanks for your interest in this detail. We have studied this in the paragraph "Metric Validity with Safeguard Models" in Sec 3. As shown by the "ASR-S (Invalid)" records in Table 3, invalid samples are relatively weaker than valid ones, while the ASR increase brought by AIA on them is even larger than that of the valid samples. That means, if these invalid samples are counted, our advantage demonstrated in the main experiments would further increase, which rules out the possibility that excluding invalid samples introduces any potential over-estimation of our method.

---

> > ### Author Rebuttal · Reviewer_dpi2 · 2026-04-04
> >
> > The internal experiments along with the additionally provided results on 'I'm happy to' have fully addressed my concerns. Should these findings be incorporated into the paper, they would substantially strengthen the paper's central claims.
> > However, the absence of qualitative textual examples of Invalid responses remains an open question. I would recommend that this limitation be explicitly acknowledged in the Limitations section or the Appendix of the camera-ready version.

---

> > > ### Author Response · Authors · 2026-04-04
> > >
> > > Thank you very much for your detailed feedback and insightful follow-up question! We fully understand your point regarding the invalid responses. Still, we would like to have a brief explanation. First, we are glad to add some qualitative examples on both sides in Appendix, while it might not be very intuitive to human readers the subtle difference between them, as the only systematic difference here lie on the judge LLM. Second, as we adopt the same standard of the judge LLM from PAIR, which is widely recognized and also adopted by multiple baselines, we respectfully believe this concern also holds for them, and its solution might beyond our scope. Finally, by the aforementioned content in Sec 3 and Table 3, our work has actually gone into deeper on this point than previous works. We sincerely hope we could be credited rather than punished by doing this. Of course, we really appreciate your kind recommendation, and are glad to add an explicit discussion in the Limitations section. Thank you!

---

### Official Review · Reviewer_fyAf · 2026-03-13

**Soundness:** 3
**Presentation:** 3
**Significance:** 3
**Originality:** 3
**Overall Recommendation:** 4
**Confidence:** 2

**Summary:**

This paper studies a new jailbreak paradigm for LALMs. Instead of embedding harmful content into audio, it argues that semantically benign but acoustically stylized audio can interfere with safety alignment and help malicious text bypass refusal. Based on this observation, the paper defines Acoustic Latent Semantics, builds an interference-audio arsenal, and proposes Acoustic Interference Attack, a universal, optimization-free, black-box attack. Experiments on 10 LALMs show that AIA substantially improves attack success over text-only baselines, and the paper further provides comparisons with prior audio jailbreak methods, safeguard-model validation, human consistency checks, and mechanistic analyses of refusal-logit suppression and latent drift.

**Compliance With Llm Reviewing Policy:**

Affirmed.

**Final Justification:**

The rebuttal addressed my main concerns. I maintain my assessment and support weak acceptance.

**Key Questions For Authors:**

Please refer to the weaknesses listed above.

**Limitations:**

yes

**Strengths And Weaknesses:**

Strengths：
1. The paper explores an interesting and timely problem: whether paralinguistic acoustic cues alone can weaken safety alignment in LALMs. This framing is novel and practically relevant.
2. The empirical study is fairly extensive: it covers multiple open-source and proprietary LALMs, compares against text-only baselines, includes prior jailbreak baselines, and provides both safeguard-model and human-consistency checks.
3. The overall narrative is clear.

Weaknesses：
1. The proposed ALS arsenal is constructed solely from Bark-generated audio. This raises a potential limitation regarding generator-specific bias: it is unclear whether the discovered interference patterns reflect a general acoustic vulnerability of LALMs or are partially tied to the stylistic manifold of Bark. It would strengthen the paper to evaluate whether combining multiple audio generation models yields a more diverse and more transferable interference arsenal.
2. Some key claims, such as “inference path drift” as the central explanation, are suggestive and interesting, but still not fully conclusive; the interpretability section supports the hypothesis, though perhaps not strongly enough to establish it as the definitive mechanism.

---

> ### Author Rebuttal · Authors · 2026-03-30
>
> Thank you for your comments. We appreciate your encouragement of our novelty, solidity, and narrative. We hope the following response would be found useful in addressing your concerns :)
>
> ---
>
> **W1: Bark as surrogate only**
>
> _\* We would like to respond to you and Reviewer zZyL together here, as both of you mentioned this question. Some of the points below might go a little beyond your comment (but still about solving Bark-specific bias). Please forgive us for this, as our response space is very limited. Thank you!_
>
> Thank you both for pointing this out. We agree with you. Just as an explanation (rather than an argument), we utilize Bark because its unique "history prompt" includes a continuous high-dimensional embedding containing the information only for audio style, thus it would be practically easy to collect the intrinsic distribution of ALS (without being disturbed by other irrelevant information like semantic content) simply via Bark unconditional sampling.
>
> We are glad to add a preliminary experiment with Tortoise TTS (thanks to the recommendation from Reviewer zZyL) on JBB data and Qwen2.5-Omni, GPT-4o-Audio models, where we set "voice=random" for Tortoise to randomly sample latent vectors. The Tortoise-based ASR and query time results are as follows:
>
> |                 | ASR-Text | ASR-Bark | ASR-Tortoise | Query-Bark | Query-Tortoise |
> |-----------------|:--------:|:--------:|:------------:|:----------:|:--------------:|
> | Qwen2.5-Omni |   50.98  |  100.00  |     98.04    |    9.67    |      10.74     |
> | GPT-4o-Audio    |   56.10  |   75.61  |     76.60    |    10.68   |      10.22     |
>
> Since Tortoise TTS utilizes a completely different diffusion-based decoder, this result serves as empirical evidence that the effectiveness of ALS is indeed universal. Due to the limited time, we would cover more data/models and other recommended TTS, like VALL-E and XTTS, in the future revision of our manuscript.
>
> Next, regarding whether Bark artifacts in ALS drive jailbreak via encoder confusion rather than paralinguistic interference, we sincerely believe Sec 4.2 has eliminated such a possibility. As shown in Fig. 6 and Fig. 7, strict statistical analysis has demonstrated that the top-3 impactful dimensions are respectively "Persona-Pitch", "Delivery-Valence", and "Delivery-Speed", while "Signal-Glitch/Artifacts" only shows a moderate level of influence.
>
> Finally, we respectfully believe it is not necessary to directly calculate the overlap between the latent space of Bark and audio encoders, because our main experiment has included basically all the mainstream LALMs, which adopt potentially different encoders, while our method shows consistent effectiveness across them, which has already demonstrated its generalization ability w.r.t. different audio encoders.
>
> ---
>
> **W2: Explanation/interpretability is not fully conclusive/definitive**
>
> Thanks for the comment. We understand your point, and acknowledge that, compared with works directly in the field of interpretability, our interpretability section might not be that strong. Still, we respectfully believe that our work is the very first step towards the new "audio interference" direction, and our interpretation is mainly to provide some preliminary mechanistic discussion (e.g., the "inference path drift") to facilitate better understanding and potential follow-up explorations. We are glad to see you find it "suggestive and interesting", which implies our goal is largely achieved. Besides, except for the "inference path drift", we sincerely think the general interpretation of the 12-dimensional ALS audio patterns might also be found insightful by the community, even when exploring other technical directions for stronger audio jailbreaks.

---

> > ### Author Rebuttal · Reviewer_fyAf · 2026-04-03
> >
> > We thank the authors for the detailed response, and I maintain my positive score.

---

> > > ### Author Response · Authors · 2026-04-03
> > >
> > > Thank you for your kind feedback! We appreciate your positive recommendation.

---

### Decision · Program_Chairs · 2026-04-30

**Decision:**

Accept (regular)

**Comment:**

This paper proposes Acoustic Interference Attack (AIA), a new jailbreak paradigm for Large Audio Language Models in which the audio modality serves not as a carrier of malicious content, but as an interference signal that disrupts safety alignment. The method constructs an acoustic latent semantic arsenal of benign-content interference audio and uses it as a universal trigger paired with malicious text prompts. Experiments on 10 LALMs across multiple datasets show strong attack effectiveness, and the paper further provides safeguard-model validation, human consistency checks, and interpretability analyses.

The paper addresses an important and timely problem in the security of LALMs. Reviewers found the central idea novel and practically relevant, especially the insight that non-semantic acoustic patterns alone can weaken safety alignment without embedding harmful content into the audio itself. The paper received four Weak Accept recommendations. During the rebuttal, the authors addressed the main concerns by adding evidence beyond the Bark-based setting, clarifying the role of sentence content versus acoustic interference, strengthening the discussion of metric validity and baseline comparison, and improving the positioning of the interpretability claims. While some limitations remain, particularly regarding the strength of the mechanistic explanation and the breadth of unified baseline comparisons, the reviewers generally agreed that the rebuttal strengthened the paper and supported acceptance.

The AC concurs with the positive reviewer consensus. Overall, the paper offers a meaningful and timely contribution to audio jailbreak research, and its strengths outweigh the remaining weaknesses. Therefore, the AC recommends acceptance to ICML 2026.